# Place Classification Algorithm Based on Semantic Segmented Objects

**Woon-Ha Yeo, Young-Jin Heo, Young-Ju Choi and Byung-Gyu Kim \***

Department of IT Engineering, Sookmyung Women's University, 100 Chungpa-ro 47gil, Yongsan-gu, Seoul 04310, Korea; wh.yeo@ivpl.sookmyung.ac.kr (W.-H.Y.); yj.heo@ivpl.sookmyung.ac.kr (Y.-J.H.); yj.choi@ivpl.sookmyung.ac.kr (Y.-J.C.)

\* Correspondence: bg.kim@sookmyung.ac.kr; Tel.: +82-2-2077-7293

**Abstract:** Scene or place classification is one of the important problems in image and video search and recommendation systems. Humans can understand the scene they are located, but it is difficult for machines to do it. Considering a scene image which has several objects, humans recognize the scene based on these objects, especially background objects. According to this observation, we propose an efficient scene classification algorithm for three different classes by detecting objects in the scene. We use pre-trained semantic segmentation model to extract objects from an image. After that, we construct a weight matrix to determine a scene class better. Finally, we classify an image into one of three scene classes (i.e., indoor, nature, city) by using the designed weighting matrix. The performance of our scheme outperforms several classification methods using convolutional neural networks (CNNs), such as VGG, Inception, ResNet, ResNeXt, Wide-ResNet, DenseNet, and MnasNet. The proposed model achieves 90.8% of verification accuracy and improves over 2.8% of the accuracy when comparing to the existing CNN-based methods.

**Keywords:** scene/place classification; semantic segmentation; deep learning; weighting matrix; convolutional neural network

## 1. Introduction

The scene is an important information which can be used as a metadata in image and video search or recommendation systems. This scene information can provide more detailed situation information with time duration and character who appears in image and video contents.

While humans naturally perceive the scene they are located, it is a challenging work for machines to recognize it. If the machines could understand the scene they are looking, this technology can be used for robots to navigate, or searching a scene in video data. The main purpose of scene classification is to classify name of scenes of given images.

In the early days, scene or place classification was carried out through traditional methods such as Scale-Invariant Feature Transformation (SIFT) [1], Speed-Up Robust Features (SURF) [2], and Bag of Words (BoW) [3]. In recent years, deep learning with the convolutional neural networks (CNNs) has been widely used for image classification ever since AlexNet [4] won the ImageNet Large Scale Visual Recognition Competition (ILSVRC) in 2012.

There have been several approaches to classify scenes and dplaces. One approach is using classification method such as *k*-nearest neighbor (KNN) classifier and other is based on the convolutional neural networks (CNNs). Chowanda et al. [5] proposed a new dataset for image classification and experimented with their dataset with CNNs such as VGG, GoogLeNet to classify Indonesian regions. Raja et al. [6] proposed a method of classifying indoor and outdoor by using KNN classifier. Viswanathan et al. [7] suggested an object-based approach. However, their methods could

classify only indoor scenes such as kitchen, bathroom, and so forth. Yiyi et al. [8] also proposed an object-based classification method combining CNN and semantic segmentation model and classified five indoor scenes. Zheng et al. [9] suggested a method for aerial scene classification by using pre-trained CNN and multi-scale pooling. Liu et al. [10] proposed a Siamese CNN for remote sensing scene classification, which combined the identification and verification models of CNNs. Pires et al. [11] analyzed a CNN model for aerial scene classification by transfer learning. These methods were focused on verifying the aerial scene.

In this paper, we propose a method for classifying a scene and place image into one of three major scene categories: indoor, city, and nature which is different from the previous works in that we classify outdoor as well. There are many objects in the scene and place. It means that we are able to classify the scene by utilizing the information of the existing objects. Also, when humans see a scene or place, they recognize the scene or place based on objects, especially background objects. If there are mountains and the sky in the scene, they would perceive it as a natural scene, and if the scene is full of buildings, they would consider it as an urban scene. If there are ceiling and walls, they would recognize it as an indoor environment.

In order to classify a scene or place image based on this human perception process, we first conduct object segmentation using the image segmentation model pre-trained with MS COCO-stuff dataset [12]. While MS COCO dataset [13] is for object detection, MS COCO-stuff dataset is for segmentation. In this paper, we used DeepLab v2 [14] model which is semantic segmentation model and can be trained with the background (stuff) objects. MS COCO-stuff dataset contains 171 kinds of object classes which are suitable for our solution. To classify an image, we construct a weight matrix of each object classes so that we can give more weight to objects that are more dominant on determining a scene. Finally, we classify the scene by combining the weight matrix and detected objects in the scene or place.

We organize the rest of this paper as follows: Section 2 highlights the details of the proposed classification method. Section 3 presents the implementation of the experiment as well as the experimental results. Finally, we draw the conclusions and suggest further research directions in Section 4.

## 2. Proposed Classification Algorithm

The overall process of the proposed classification method is summarized in Figure 1. It contains image segmentation stage and computing scene score of the image. Before carrying this process out, we design the weight matrix with size 171 × 3. This matrix consists of 171 object classes, and each object class has 3 scene labels. The details of constructing the weight matrix will be explained in the following Section 2.2.

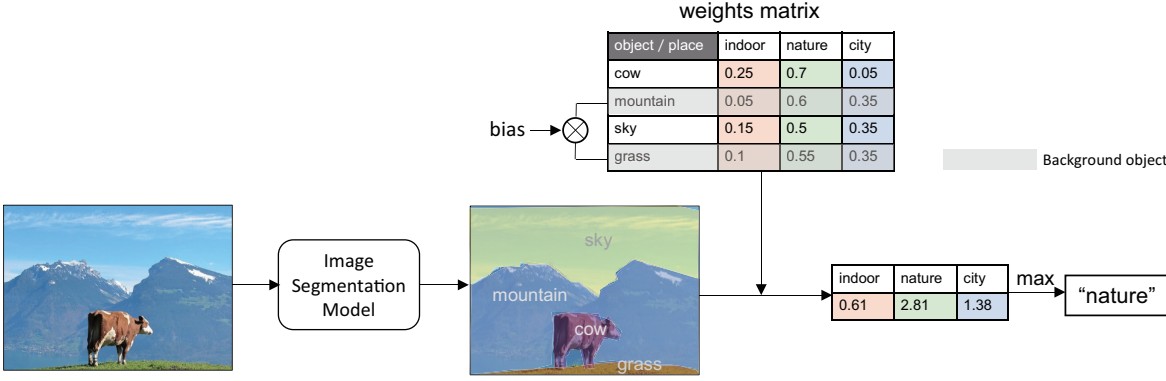

**Figure 1.** Overall structure of the proposed scheme.

### 2.1. Overall Structure

We define scene categories into three classes—indoor, city, and nature, which are the major categories for representing some scenes. To classify a scene image into a scene class, we first feed the image to the pre-trained image segmentation model. In this paper we used DeepLab v2 [14] to get the objects in the scene. For instance, if we get 'sky', 'mountain', 'cow', and 'grass' as a result of segmentation, we look for these object classes in the pre-constructed weight matrix. As humans perceive the scene in a scene image more by background objects rather than thing (foreground) objects, we want to focus more on the background classes by giving more weight on these classes. Therefore, if detected objects are part of background classes, we multiply *bias* value of 1.4 which is determined empirically.). We compute the scores of scene and place classes of the image by adding all weights of objects in the image. The scene class with the highest score is determined as the final scene and place class of the image. We determine objects in ground, solid, building in outdoor tree (Figure 2) as background objects.

### 2.2. Design of Weighting Factors

We build the weight matrix shown in Figure 3 by using 2017 validation images from the COCO-stuff dataset, including 5K images. These images are manually labeled as scene classes. After preparing dataset for constructing the matrix, the images are fed to the pre-trained image segmentation model one by one. We can get one or more objects as a result of segmentation.

The COCO-stuff dataset [12] includes 80 "thing" classes and 91 "stuff" classes, and stuff classes are divided into two wide categories—indoor and outdoor. The outdoor classes contain various classes representing background of an image such as building, mountain, and so forth (Figure 2).

The weight matrix $W$ has size of $M \times N$, $M$ is the number of object classes, and $N$ is the number of scene classes. Since we use COCO-stuff dataset and three scene labels, it turns out to $171 \times 3$. It is initialized with zeros at first.

As shown in Figure 3, assuming that we get classes of 'cow', 'mountain', 'sky', and 'grass' from an image. The image has a nature scene, so add 1 to nature column in weight matrix for each object class (cow, mountain, sky, and grass). After iteration of this process under 5 K images, the matrix would be constructed with various numbers. Therefore, we normalize it for each row. In the Equation (1), $W'$ denotes the normalized weight matrix. In addition, $m$ is $m$-th object class in the dataset and $n$ is $n$-th label in the place classes. The algorithm of constructing the weighting matrix is described in Algorithm 1 and the inference process with the model is shown in Algorithm 2.

In the inference process, we perform semantic segmentation for each test image, and we compute the scores of scene or place classes of the image by multiplying bias value to background object weights and adding all of them of each scene or place class by using pre-constructed weight matrix as:

$$W'_{mn} = \frac{W_{mn}}{\sum_{n'=1}^{N} W_{mn'}}.$$ (1)

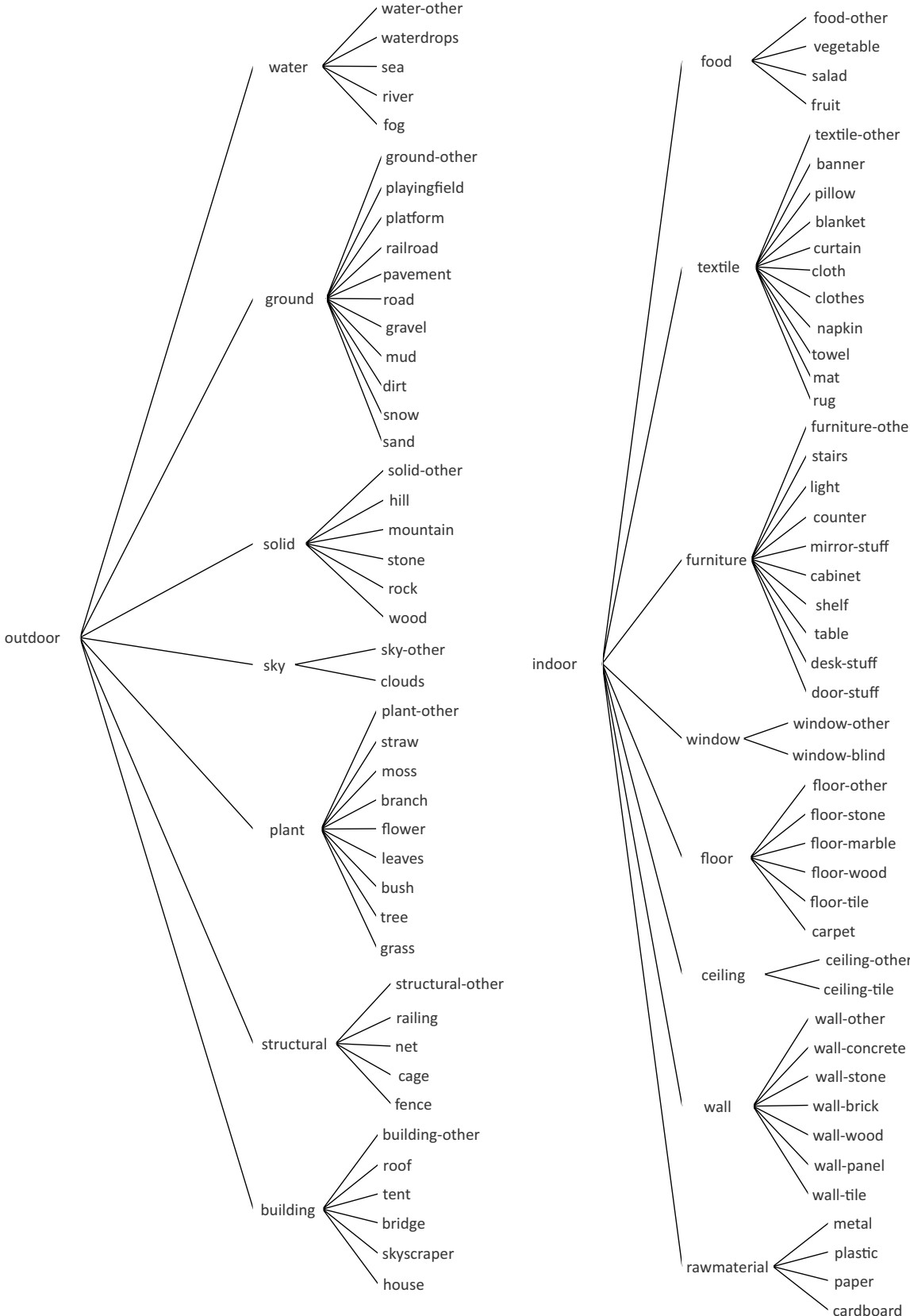

**Figure 2.** COCO-Stuff label hierarchy.

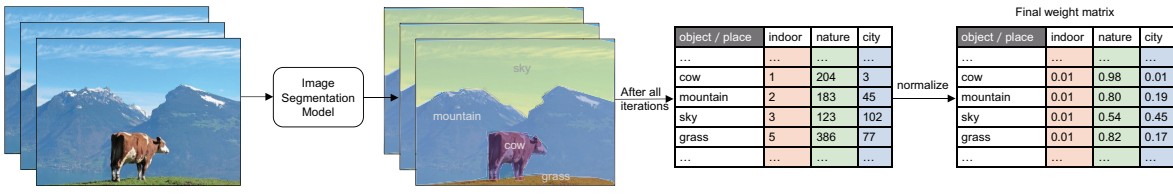

COCO-stuff validation 2017 images (5K)

**Figure 3.** Construction of weight matrix *W* with a sample image.

---

**Algorithm 1** Weighting Matrix *W*

---

**Require:** *F*: pre-trained semantic segmentation model

1: **Inputs:**

    scene-labeled images $\{(x_1, y_1), (x_2, y_2), ..., (x_n, y_n)\}$

2: **Initialize:**

    $W[i][j] \leftarrow 0, i = 1, \ldots, o, j = 1, \ldots, p$

    *o*: # of object classes, *p*: # of scene classes

3: **for** t = 1 to n **do**

4:     $O_t \leftarrow F(x_t)$

5:     **for** $o \in O_t$ **do**

6:         $W[o][y_t] \leftarrow W[o][y_t] + 1$

7:     **end for**

8: **end for**

9: **for** i = 1 to o **do**

10:     **for** j = 1 to p **do**

11:         $W[i][j] \leftarrow W[i][j] / (W[i][1] + \ldots + W[i][p])$

12:     **end for**

13: **end for**

---

---

**Algorithm 2** Inference Process

---

**Require:** *F*: pre-trained semantic segmentation model and $W[i][j]$: weight matrix

1: **Inputs:**

    Test images $\{x_1, x_2, ..., x_n\}$

2: **Initialize:**

    $V[i] \leftarrow 0, i = 1, \ldots, p$

    *o*: # of object classes, *p*: # of scene classes

3: **for** t = 1 to n **do**

4:     $O_t \leftarrow F(x_t)$

5:     **for** $o \in O_t$ **do**

6:         $bias \leftarrow 1$

7:         **if** o is in Background **then**

8:             $bias \leftarrow 1.4$

9:         **end if**

10:         $V[t] \leftarrow V[t] + W[o] \times bias$

11:     **end for**

12: **end for**

13: $\hat{y} \leftarrow argmax(V[t])$

---

## 3. Experimental Results and Discussion

In this section, we will show the results of our classification model and well-renowned classification methods using CNNs. We implemented the proposed scheme by using PyTorch deep learning framework, and used single GPU for training. We trained DeepLab v2 model with COCO-stuff

dataset which contains total 164k images. As we mentioned it in Section 2.2, the reason why we use COCO-stuff dataset is because its object classes are divided into indoor and outdoor categories.

In order to compare our method with CNN based classification models, we first built a custom test dataset that consists of 500 images as shown in Table 1. We extracted them from Korean movies, and the images were manually labeled into three scene classes (i.e., 0: indoor, 1: nature, 2: city) under certain criteria. We set some criteria according to the logic that humans more focus on the background objects rather than foreground objects. The criteria are described in Figure 4.

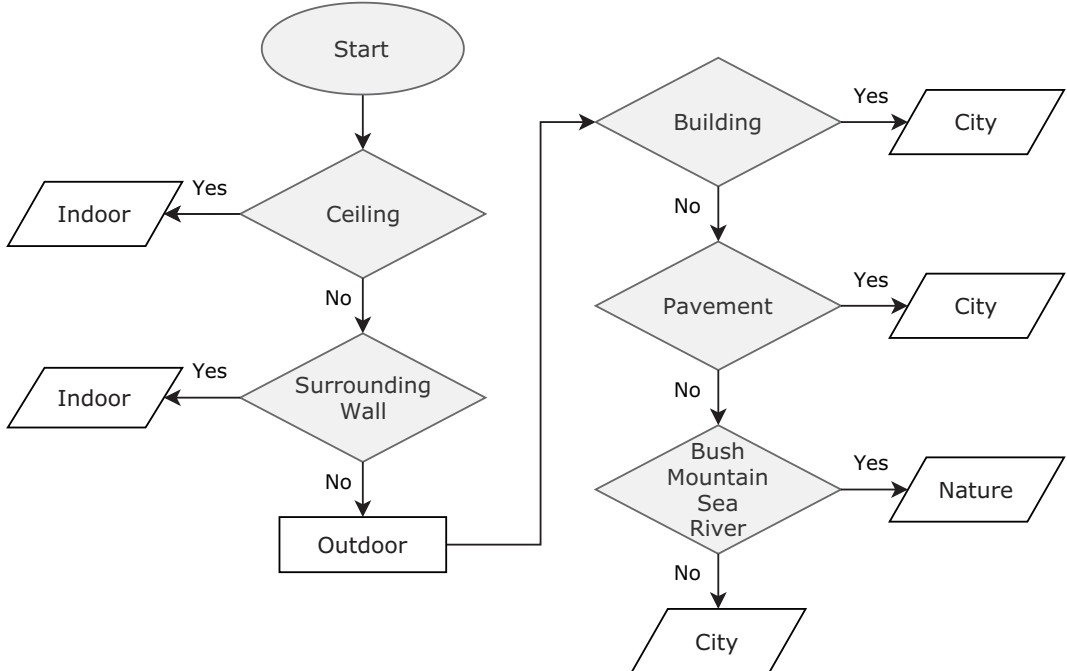

**Figure 4.** The criteria of labeling custom dataset. We first label the scene as either indoor or outdoor according to the existence of ceiling and surrounding wall. Then city images are labeled by buildings and pavements which are usually existing in urban area. Lastly, the scene is labeled as nature when there is nature stuff such as bush, mountain, sea, or river.

The sample test images are shown in Figure 5. We used COCO-stuff validation 2017 dataset for training the CNN models, which were also used for building the weight matrix. The test images were used for measuring accuracy of the classification. We experimented various CNNs, such as VGG [15], Inception [16,17], ResNet [18], ResNeXt [19], Wide-ResNet [20], DenseNet [21], and MnasNet [22] as shown in Table 2.

To be more specific, we trained each model by using transfer learning scheme [23], especially Feature Extraction [24,25]. The basic concept of feature extraction is represented in Figure 6. Typical CNNs have convolutional layers for extracting good features and fully connected layers to classify the feature. Feature extraction technique which trains only fully connected layers is used when there are insufficient data for training.

PyTorch Deep Learning Framework was used again to implement all structures for the experiment. The results of accuracy were computed after 200 iterations of training. We trained each model using cross entropy loss, and Adam optimizer with a batch size 64, learning rate 0.001. Learning rate was multiplied by 0.1 every 7 iteration.

In Table 2, we can observe the proposed scheme outperforms the existing well-known CNNs which were trained using transfer learning. In terms of the accuracy, the proposed scheme achieved 90.8% of the accuracy.

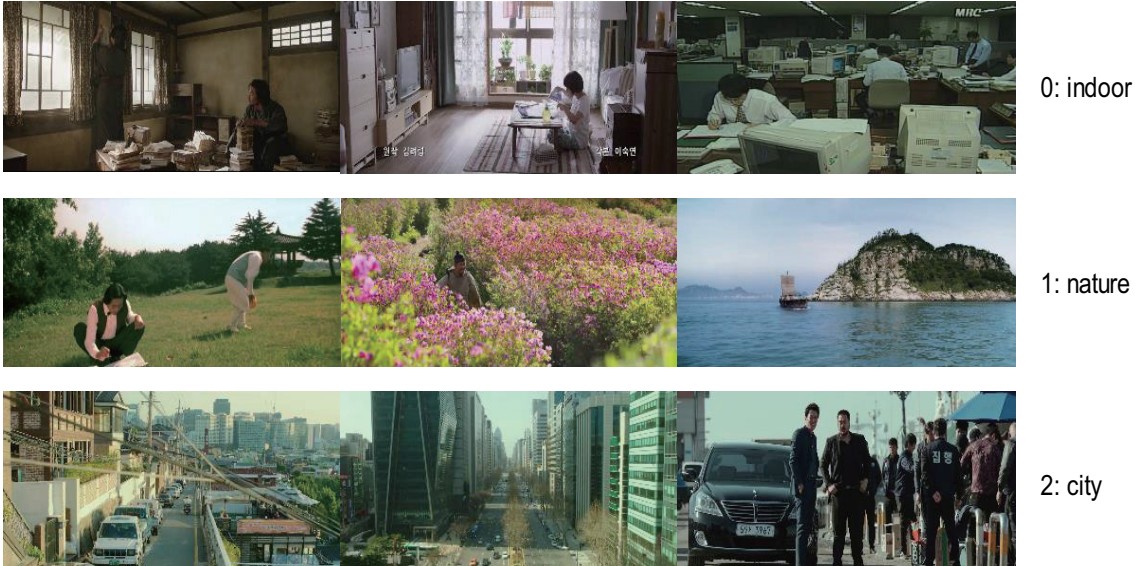

**Figure 5.** Samples of custom test dataset; each row represents each scene class.

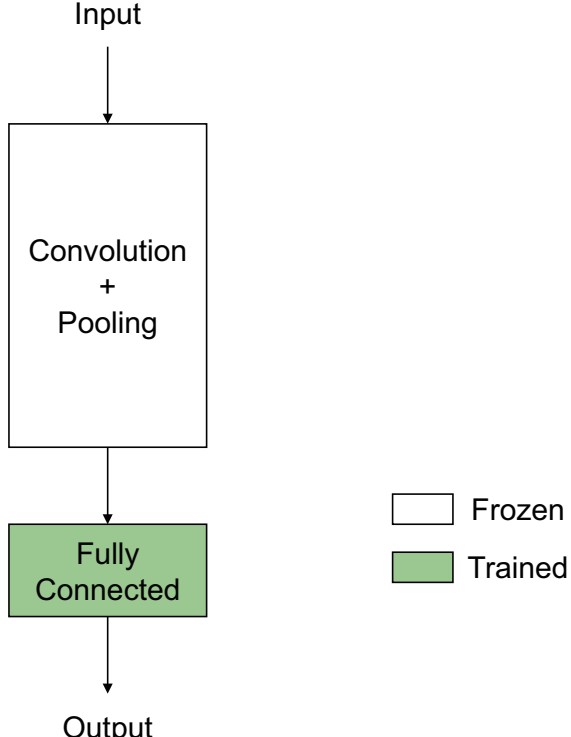

**Figure 6.** Basic Concept of Feature Extraction. The white box represents convolution and pooling layers in convolutional neural networks (CNNs), which are used for extracting features of an image. This part is not trained during transfer learning process. The green box represents fully connected layer in CNNs, which operates as classifier. This part is trained during whole training process.

**Table 1.** The number of images for each class in our custom dataset.

| Indoor | Nature | City | Total |
|--------|--------|------|-------|
| 305    | 106    | 89   | 500   |

**Table 2.** Performance comparison with the existing CNNs on indoor/nature/city classes.

| Models | Accuracy (%) |
| --- | --- |
| VGG-19 [15] | 79.6 |
| VGG-19 (BN) [15] | 88.0 |
| GoogLeNet [16] | 84.6 |
| Inception-v3 [17] | 86.2 |
| ResNet-101 [18] | 84.0 |
| ResNeXt-101 [19] | 88.0 |
| Wide-ResNet [20] | 85.2 |
| DenseNet-121 [21] | 84.0 |
| DenseNet-161 [21] | 83.6 |
| DenseNet-201 [21] | 85.2 |
| MnasNet (0.5) [22] | 76.6 |
| MnasNet (1.0) [22] | 79.8 |
| Proposed Method | 90.8 |

When compared with VGG-19 (BN) [15] and ResNeXt-101 [19], the proposed method could improve 2.8% of the accuracy. Also, our scheme improved the performance over 13% comparing to MnasNet (0.5) [16]. VGG-19 was tested with batch normalization (BN) and without BN. The float values with MnasNet is the depth multiplier in Reference [22]. From this result, we can see that the proposed scheme is very reliable and better to classify the scene.

Figure 7 represents the graph of the experiment on determining optimal bias value in terms of test accuracy. It shows that the highest test accuracy when the bias is 1.4. This value is used in the inference process when multiplying weights of background objects.

In addition, we measured test accuracy on COCO-Stuff test dataset. We adopted first 100 images of the test images and labeled the images according to the criteria in Figure 4. We used same parameters as the previous experiment while training CNNs and building weight matrix. The result is shown in Table 3. The result shows that the proposed method outperforms the conventional CNNs and also indicates that it achieves better performance when test images are taken in the same domain with train images.

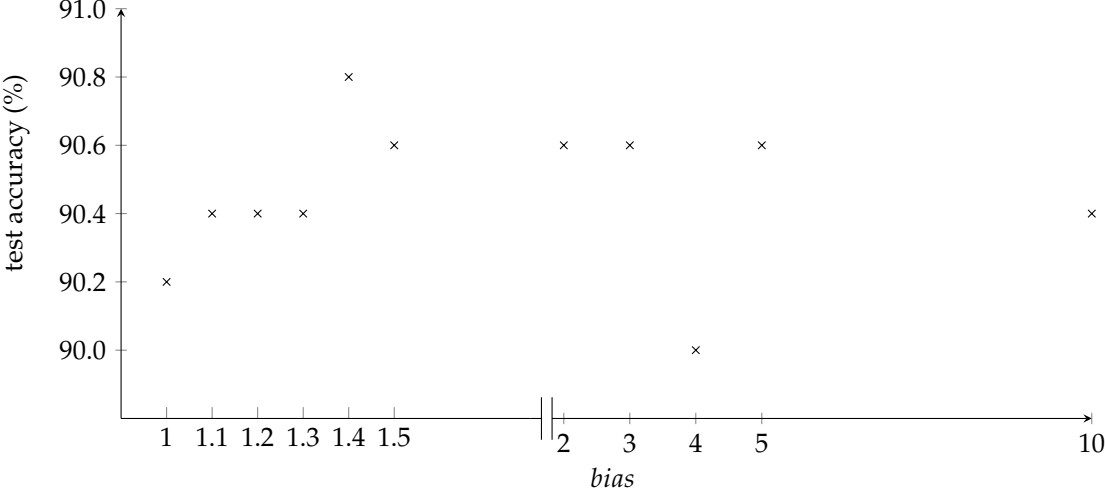

**Figure 7.** Experiment on different bias values for multiplying weights of background objects when inferencing.

**Table 3.** Performance comparison with the existing CNNs using COCO-Stuff test set.

| Models | Accuracy (%) |
|---|---|
| VGG-19 [15] | 91.0 |
| VGG-19 (BN) [15] | 85.0 |
| GoogLeNet [16] | 88.0 |
| Inception-v3 [17] | 86.0 |
| ResNet-101 [18] | 89.0 |
| ResNeXt-101 [19] | 89.0 |
| Wide-ResNet [20] | 88.0 |
| DenseNet-121 [21] | 82.0 |
| DenseNet-161 [21] | 86.0 |
| DenseNet-201 [21] | 90.0 |
| MnasNet (0.5) [22] | 87.0 |
| MnasNet (1.0) [22] | 85.0 |
| Proposed Method | 92.0 |

Lastly, we experimented on indoor classes and Table 4 shows the results. We used the subset of MITPlaces dataset [26]. It contains categories of 'library', 'bedroom' and 'kitchen' and 900 train images and 100 test images per class. As previous experiment, train images are used for building weight matrix and test images are used for measuring test accuracy in the proposed method. Classifying indoor categories must be treated differently from classifying indoor and outdoor. Since all indoor scenes have ceilings and walls, the bias value in Algorithm 2 must be given not by background objects, but by foreground objects. In this experiment, we defined foreground objects as furniture categories in Figure 2 and determined the value to be 3 empirically. Although the results shows that 7 of CNNs outperforms our method by less than 4.6%, it shows that our method can be extendable to indoor categorization problem.

**Table 4.** Performance comparison with the existing CNNs on library/bedroom/kitchen classes.

| Models | Accuracy (%) |
|---|---|
| VGG-19 [15] | 92.0 |
| VGG-19 (BN) [15] | 94.0 |
| GoogLeNet [16] | 88.0 |
| Inception-v3 [17] | 93.7 |
| ResNet-101 [18] | 91.0 |
| ResNeXt-101 [19] | 95.3 |
| Wide-ResNet [20] | 88.0 |
| DenseNet-121 [21] | 93.0 |
| DenseNet-161 [21] | 94.3 |
| DenseNet-201 [21] | 90.3 |
| MnasNet (0.5) [22] | 89.7 |
| MnasNet (1.0) [22] | 90.0 |
| Proposed Method | 90.7 |

CNNs which showed the best performance tested with our custom dataset are VGG-19 (BN) and ResNeXt-101. They both showed test accuracy of 88% and the proposed method showed 90.8%. Table 5 represents the performance of three models on each of three scene classes. VGG-19 (BN) predicted all images perfectly in indoor class and the proposed method is following. In nature class, the proposed method showed the best accuracy. When it comes to city class, ResNext-101 showed the best results. From this result, we can see that the proposed method is reliable for scene classification. Source code is available at https://github.com/woonhahaha/place-classification.

**Table 5.** Comparison between best CNNs and the proposed method. The correct number of images per class in our custom dataset.

|  | Indoor | Nature | City | Total |
|---|---|---|---|---|
| VGG-19 (BN) | 305 | 79 | 56 | 440 |
| ResNeXt-101 | 282 | 88 | 70 | 440 |
| Proposed Method | 304 | 89 | 61 | 454 |

## 4. Conclusions

In this work, we have proposed an efficient scene and place classification scheme using background objects and the designed weighting matrix. We designed this weighting matrix based on the open dataset which is widely used in the scene and object classifications. Also, we evaluated the proposed classification scheme which was based on semantic segmentation comparing to the existing image classification methods such as VGG [15], Inception [16,17], ResNet [18], ResNeXt [19], Wide-ResNet [20], DenseNet [21], and MnasNet [22]. The proposed scheme is the first approach of object-based classification that can classify outdoor categories as well. We have built a custom dataset of 500 images for testing which can help researchers who are dealing with scene classification. We crawled frames from Korean movies and labeled each image manually. The images were labeled as three major scene categories (i.e., indoor, nature, and city).

Experimental results showed that the proposed classification model outperformed several well-known CNNs mainly used for image classification. In the experiment, our model achieved 90.8% of verification accuracy and improved over 2.8% of the accuracy when comparing to the existing CNNs.

The Future work is to widen the scene classes to classify not just indoor (library, bedroom, kit) and outdoor (city, nature), but also more subcategories. It would be helpful for searching in videos with such semantic information.

**Author Contributions:** Conceptualization, B.-G.K.; methodology and formal analysis, W.-H.Y.; validation, Y.-J.C. and Y.-J.H.; writing–original draft preparation, W.-H.Y. and Y.-J.C.; writing–review and editing, B.-G.K.; supervision, B.-G.K. All authors have read and agreed to the published version of the manuscript.

**Funding:** This research project was supported by Ministry of Culture, Sports and Tourism (MCST) and from Korea Copyright Commission in 2020.

**Acknowledgments:** Authors thank for all reviewers who helped to improve this manuscript.

**Conflicts of Interest:** The authors declare no conflict of interest.

## Abbreviations

The following abbreviations are used in this manuscript:

| | |
|---|---|
| CNN | Convolutional Neural Network |
| BN | Batch Normalization |

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
