# Peer review of "Place Classification Algorithm Based on Semantic Segmented Objects"

_applsci, doi:10.3390/app10249069_

Round 1
Reviewer 1 Report
This paper proposes a method for scene classification. The method is based on the use of an object recognition CNN (DeepLab). From the objects detected, the method applies a classification matrix with weights from any object to any scene.
The review of the state of the art must be extended. There is only a small (and incomplete) review in the Introduction section. More scene classification refereces, using the whole image and using objects, must be included.
The comparison between the proposed method and the SOTA is not fair. By one hand, the methods compared with are CNNs, trained with a given dataset (COCO). But test images were taken from other domain (movies). I wonder what is the result if test images were taken from the COCO dataset (images not used in training). By the other hand, the robustness of the proposed method comes from the robustness of DeepLab: if DeepLab is able to detect the objects in the image, then the proposed method works well.
I would like to see a comparison with other state-of-the-art methods (the ones missed in the review).
I would also see some experiments with indoor scenes. And with more classes, not just three. I do not see clearly if the proposed method is extendable.
Author Response
The authors would like to express their sincere thanks to the reviewers
for their good comments.
Please, refer to the attachment for replies.
Thank you very much.

Reviewer 2 Report
This manuscript compares DeepLab-v2 trained on COCO-Stuff, the 171 segmentation classes are then mapped on 3 classes (nature,indoor,city) using a weight matrix. This matrix is manually constructed and not learned. It is then compared to a feature extraction strategy applied to various pre-trained CNN. The final classifiers are then re-trained using some training data but it is not clear what is this dataset. The new contribution is marginal in my opinion, the comparisons between models is difficult to interpret given that they are trained on different datasets, a fine tuning approach is not considered here also.
Author Response

(The authors gave the same response as above.)

Reviewer 3 Report
The Authors propose a new scene classification algorithm based on a weighting matrix which assigns more weight to objects more dominant on a scene. The matrix is designed using the image segmentation model pre-trained with MS COCO-stuff dataset. The performance of the new method on a custom test dataset outperforms classification using convolutional neural networks.
Remarks:
- Algorithm 1, line 6 why oq?
- Algorithm 2, please move the line bias <--1 down and place it before if.
- Capture for Figure 1: add that grass and mountain are here considered as background.
- Figure 3: the element “normalize” may be understood incorrectly - normalization takes place at the end.
- The custom test dataset is unbalanced. Accuracy improvement 2.8 % means 14 images. Please show how good are the best CNNs compared to your method in recognition of images from each of the three scene classes. Is the dataset publicly accessible?
Author Response

(The authors gave the same response as above.)

Reviewer 4 Report
NA
Author Response

(The authors gave the same response as above.)

Round 2
Reviewer 1 Report
Authors have fulfilled all my previous requirements.
Author Response
[1] Authors have fullled all my previous requirements.
Reply: Thank you very much for your eort to improve this manuscript.
(Also you can refer to the attachment for changed part!)

Reviewer 2 Report
The weight matrix should be made available (on GitHub for instance) in order to ensure that other people can replicate your work. Why there is no comparisons with DeepLab-v2 used as a deep feature classifier also?
Author Response
[1] The weight matrix should be made available (on GitHub for instance)
in order to ensure that other people can replicate your work.
Reply: Thank you for your kind comment. We have added the
url of source code in Abstract section of the revised manuscript.
After: (p. 7, lines 162-163)
Source code is available at https://github.com/woonhahaha/place-
classication.
[2] Why there is no comparisons with DeepLab-v2 used as a deep feature
classier also?
Reply: Thank you for your comment. The DeepLab-v2 is a model
for the task of semantic image segmentation, not for image classi-
cation. Therefore, it is dicult to compare the performance directly.
We have just utilized the DeepLab-v2 model, which was pre-trained
with MS COCO-stu 164k train dataset, to get foreground and back-
ground objects in a scene image.
(Also you can refer to the attachment for replies...!!)

Round 3
Reviewer 2 Report
Thank you for your response.